# The use of social robots with children and young people on the autism spectrum: A systematic review and meta-analysis

Athanasia Kouroupa[1,2], Keith R. Laws[1], Karen Irvine[1], Silvana E. Mengoni[1], Alister Baird[2], Shivani Sharma[1] *

1 School of Life and Medical Sciences, University of Hertfordshire, Hatfield, United Kingdom, 2 Division of Psychiatry, University College London, London, United Kingdom

* s.3.sharma@herts.ac.uk

## Abstract

### Background

Robot-mediated interventions show promise in supporting the development of children on the autism spectrum.

### Objectives

In this systematic review and meta-analysis, we summarize key features of available evidence on robot-interventions for children and young people on the autism spectrum aged up to 18 years old, as well as consider their efficacy for specific domains of learning.

### Data sources

PubMed, Scopus, EBSCOhost, Google Scholar, Cochrane Library, ACM Digital Library, and IEEE Xplore. Grey literature was also searched using PsycExtra, OpenGrey, British Library EThOS, and the British Library Catalogue. Databases were searched from inception until April (6th) 2021.

### Synthesis methods

Searches undertaken across seven databases yielded 2145 articles. Forty studies met our review inclusion criteria of which 17 were randomized control trials. The methodological quality of studies was conducted with the Quality Assessment Tool for Quantitative Studies. A narrative synthesis summarised the findings. A meta-analysis was conducted with 12 RCTs.

### Results

Most interventions used humanoid (67%) robotic platforms, were predominantly based in clinics (37%) followed home, schools and laboratory (17% respectively) environments and targeted at improving social and communication skills (77%). Focusing on the most common outcomes, a random effects meta-analysis of RCTs showed that robot-mediated

**Data Availability Statement:** All relevant data are within the paper and its Supporting Information files.

**Funding:** The author(s) received no specific funding for this work.

**Competing interests:** The authors have declared that no competing interests exist.

interventions significantly improved social functioning (g = 0.35 [95%CI 0.09 to 0.61; k = 7). By contrast, robots did not improve emotional (g = 0.63 [95%CI -1.43 to 2.69]; k = 2) or motor outcomes (g = -0.10 [95%CI -1.08 to 0.89]; k = 3), but the numbers of trials were very small. Meta-regression revealed that age accounted for almost one-third of the variance in effect sizes, with greater benefits being found in younger children.

## Conclusions

Overall, our findings support the use of robot-mediated interventions for autistic children and youth, and we propose several recommendations for future research to aid learning and enhance implementation in everyday settings.

## PROSPERO registration

Our methods were preregistered in the PROSPERO database (CRD42019148981).

## Introduction

With an ongoing focus on early interventions [1, 2], the functioning of children and young people on the autism spectrum has progressively improved [3]. An array of interventions exists to support the development of social, emotional and life skills in autistic children but with mixed evidence of clinical effectiveness [4]. This makes it important to continue to search for approaches that are adaptable to individual needs given the heterogeneity of the autism spectrum, and scalable to advance the most benefit.

Amongst the plethora of interventions, robots have emerged as a promising aid in the development of everyday skills and as a mechanism to improve quality of life [5, 6]. Recent studies show that robots are well-accepted by children and young people on the autism spectrum and are linked to positive impact on imitation skills, eye-contact, joint attention, behavioural response, and repetitive and stereotyped behaviour [7, 8]. Several reviews have summarised that individuals on the autism spectrum interact more effectively with robots than humans to practice life-skills [7, 9, 10]. This advantage has been attributed to stimulation through repetition, simplified facial expressions that mimic humans, and a gradual increase in the level of challenge all acting as important scaffolds to mastering skills [9–12]. In other words, robots offer a predictable and consistent interaction pattern, which is favourable to the learning of children and young people on the autism spectrum [13–15].

So-called 'social robots' have shown advantages in educating children and youth on the autism spectrum in various domains, including: attention [6], learning [16], behavioural regulation [17, 18] and restricted and repetitive patterns of behaviours [19–21]. Social robots are described as physically embodied agents that have some (or full) autonomy and engage in social interactions with humans, by communicating, cooperating, and making decisions [22]. A research study classified robots based on their appearance as human-inspired, animal-inspired, imaginary, or manmade objects and functional robots (e.g., drones) [23]. Robot platforms benefit from the capacity to represent familiar social cues to children and young people in a controlled environment (e.g., facial features such as eyes). Technological advances have also enabled humanoid robots to represent a range of human-like functions, which is important for children on the autism spectrum, whose perceptual processing of humans and objects appears to be similar [24]. The emergence of social robots brings opportunity that innovative

technologies could further aid the development of skills in children and young people on the autism spectrum through playful activities and that such interaction might positively impact the learning process [25]. Further, rapid developments in technology mean that interventions can be more readily personalized, a salient feature given the heterogeneity of idiosyncratic difficulties in autism [26, 27].

Several intervention programmes have explored the impact of social robots on skills development, however evidence concerning their efficacy remains limited [28, 29]. Despite the potential of robots in autism training, significant gaps persist in the literature. Studies have mainly focused on reviewing the acceptability of robots to children and young people on the autism spectrum as well as therapists delivering interventions. Research has overlooked the variability of robot types used and their efficacy beyond the immediate intervention period [7, 22, 30]. In addition, important features such as the settings in which intervention is delivered, and characteristics such as the number of sessions needed to bring about meaningful benefit remains unclear. Examining if and how environment influences the efficacy of robot interventions is fundamental to enhance learning gain and to consider if any setting is more suitable to overcome the challenge of generalizing skills [30, 31]. It is important to know the outcomes targeted by robot-mediated interventions and to consider meaningful skills development in these domains to inform future directions for robotic research in autism, and importantly, applied clinical value. A recent study reviewed evidence from randomised control trials (RCTs) with children and adults on the autism spectrum, finding that most utilised humanoid robots, focusing on outcomes such as job interview skills, gesture production and recognition, social, mental, physical, and verbal skills [32]. While Salimi and colleagues [32] reported that 15/19 RCTs demonstrated positive gains in targeted skills development, the authors did not undertake meta-analysis to quantify efficacy for specific clusters of skill development and this is key if social robots are to be advanced in everyday care.

In the current systematic review and meta-analysis, we aimed to summarise the evidence-based on the use of social robots with children and young people on the autism spectrum, considering data from all study types and grey literature. The objectives of the review were to evaluate the different robot platforms that have been used with individuals on the autism spectrum, the settings in which interventions have been implemented, the role of robots within interventions, and the range of outcomes targeted for therapeutic gain. Within this, we explored any specific trends related to randomised and non-randomised studies. Further, we aimed to use meta-analysis to pool data from RCTs, widely accepted as the most rigorous study design, to assess the efficacy of interventions for specific domains of learning.

## Methods

The current systematic review and meta-analysis was completed in accordance with the Preferred Reporting Items for Systematic Reviews and Meta-Analysis (PRISMA) (S1 Table) [33]. The study protocol was preregistered on PROSPERO [CRD42019148981].

### Identification of studies

Papers were eligible for inclusion in the review if (a) participants were diagnosed on the autism spectrum using established diagnostic criteria (ICD-11, DSM-V or previous versions); (b) they were aged under 18 years; (c) the study included an intervention based on any robotic platform; (d) data were reported for at least one outcome against which to measure intervention gains. We only included primary studies and all designs were considered (e.g., randomised, case controlled, case reports). Lateral search techniques were also used to identify additional papers for inclusion in the review.

Criteria for exclusion in the review were the following: (1) lack of separate presentation of study outcomes for children on the autism spectrum; (2) individual aged over the age of 19; (3) lack of recording study procedures including number, duration and frequency of robot-mediated sessions; (4) no reference to the robot type/model; (5) commentary papers, protocols, surveys and reviews; (6) qualitative studies; (7) qualitative elements in mixed-method designs; (8) studies that were not published in English.

We searched the following databases: PubMed, Scopus, EBSCOhost, Google Scholar, Cochrane Library, ACM Digital Library, and IEEE Xplore. Grey literature was also searched using PsycExtra, OpenGrey, British Library EThOS, and the British Library Catalogue.

Search terms aligned to the following core domains: 'autism' AND 'robot' AND outcome-specific terms aligned to the concepts of social, emotions, communication, education, academic attainment, behaviour, or health (S2 Table). Databases were searched from inception until April (6th) 2021. Additional studies were identified through a manual search of the references in relevant studies.

## Study selection

Identified records were exported into Mendeley (v1.19.8) and duplicates removed. The first author (AK) screened all the titles and abstracts, and a second reviewer (AB) independently screened a random sample of 20% of the originally identified records, both using pre-determined inclusion criteria, to establish reliability for study selection. The full texts of potentially relevant papers were retrieved and independently assessed for eligibility by the first author (AK). Twenty percent of full text of the eligible studies were independently screened for eligibility by the second reviewer (AB). There was 100% agreement between reviewers in the selection of studies which met the inclusion/exclusion criteria.

## Data extraction

A pre-piloted form was used to extract data, including the following items: authors and year of publication, the study aims, design, methods, and a summary of the outcomes. We extracted characteristics of study participants such as mean age of all study participants, gender distribution, diagnostic tool, and intelligent quotient (IQ), where available. Sample ethnicity was also recorded, where available. The socioeconomic status was not recorded in any of the included studies and so these data could not be extracted.

Characteristics of the interventions were extracted across the articles and included: robot type, duration of intervention, frequency and length of sessions, location where intervention was delivered, and outcome measured. Other intervention characteristics such as the type of robot used, and intervention location were also extracted. Authors of 10 separate studies were contacted to extract information which was not clearly reported in published work (e.g., confirm diagnosis, session location, number and/or duration of a session, moderator of session delivery, mean and standard deviation per participant). Three authors shared information about their studies. Findings were summarized narratively, and where relevant, using means, standard deviations, and percentages.

## Meta-analysis

Meta-analyses were conducted using Comprehensive Meta-Analysis Version 3.0 for Windows [34]. We calculated Hedge's $g$ effect sizes (and 95% confidence intervals) for end-of-trial data comparing robot-mediated interventions and control groups in RCTs. Hedge's $g$ adjusts effect sizes according to sample size. Comparisons were made for intervention and control group at end-of-trial on the primary outcomes of: social, emotional, and motor benefits (which

emerged as the main clusters of outcomes in RCTs). All meta-analyses used a random effects approach. We classified effect sizes as small (0.2) medium (0.5) and large (0.8) according to Cohen's nomenclature. Heterogeneity was assessed using the I2 statistic, and for interpretation we followed Cochrane guidance (Higgins et al., 2019) where I2 values were identified: 0%-40% as might not be important; 30%-60% as may represent moderate heterogeneity; 50%-90% may represent substantial heterogeneity; 75%-100% representing considerable heterogeneity.

## Quality assessment

Two reviewers (NK and AB) independently measured the quality of the included studies using the Quality Assessment Tool for Quantitative Studies [35]. The assessment tool assesses six components of study validity including: selection bias, study design, confounders, blinding, data collection methods, withdrawals, and dropouts. Each component is rated as strong (1), moderate (2), or weak (3). Each paper receives an overall mark ranging between "strong (no weak rating)", "moderate (one weak rating)" and "weak (two or more weak ratings)". All studies were appraised independently by the first reviewer (NK). The second reviewer (AB) reviewed independently 20% of the included studies. The inter-rater reliability between the authors, using Cohen's Kappa was 'strong' (0.87 agreement). Any disagreements between the two reviewers were resolved through discussion or by consulting a third reviewer (SS). The results of the quality analysis were further tabulated to identify any types of bias common to the included studies (S3 and S4 Tables).

# Results

The study selection process and a summary of included articles will be presented first, followed by a general overview of the quality of research. Next, the main results will be presented according to our review objectives.

## Selection and inclusion of studies

The search generated 2145 records. After removing duplicates, 1646 records were screened, with 151 deemed relevant and full texts reviewed, of which 44 articles reporting 40 studies were deemed eligible for inclusion into the systematic review. The most common reason for exclusion was the lack of information concerning diagnostic method for autism (n = 57) followed by studies which did not meet the inclusion criteria such as reviews, protocols, surveys, feasibility trials, opinion letters (n = 23). A smaller number of studies were excluded because of the following reasons: 1. a new robotic platform/ intervention was developed (n = 9); 2. adults/ children with diagnoses other than autism were examined (n = 7); 3. the robot name/type was missing (n = 4); 4. a qualitative study had been conducted (n = 2) (Fig 1).

## Characteristics of included studies

The description of the study characteristics is based on 40 studies. Four articles [21, 36, 37] had overlapping samples and were not included in the average sample size. The majority of the studies were non-randomized (n = 23, 57%), followed by RCTs (n = 17, 43%) (Table 1). Thirty-four studies utilised video data to analyse the study findings. The average sample size of children and young people on the autism spectrum across the studies was 10 (range 1–30), (M = 10.22, SD = 6.58, k = 40).

Most studies recruited more males than females, with percentage of males ranging from 67% to 100%. Ethnicity of participants was reported in nine studies (22%), of which five studies included Chinese participants [38–42], three comprised Caucasian, African American, Asian

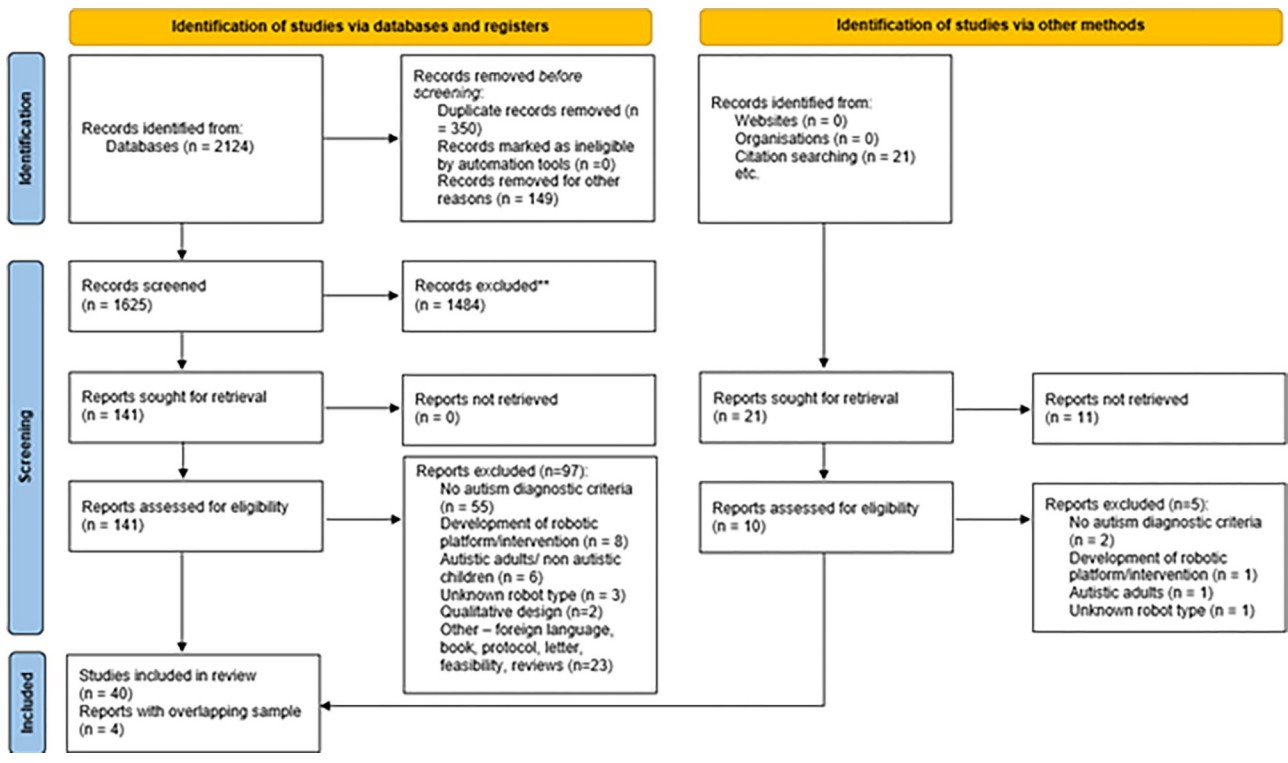

**Fig 1. PRISMA flow diagram of the study selection process.**

Hispanic, Mixed African American and Caucasian, Mixed Caucasian and Hispanic and Latino participants [20, 43, 44] and one study included Italian children [45].

Comparing RCT and non-RCT designs, the former had more complete reporting about the autism diagnosis process (for example, from which type of healthcare professional). Only sixteen studies (40%) measured the cognitive capacity (IQ) of children. Further characterisation of participants (e.g., school type, parent's demographics) was generally poor. Child/adolescent participants ranged in age from 2 years to 16 years (M = 7.4; SD = 3.08). Overall, the studies were published between 2008 and 2020, in Europe (k = 18, 41%) [i.e., Romania (k = 7), Portugal (k = 3), France (k = 2), Italy (k = 2), Netherlands (k = 2), Belgium (k = 1) and Luxembourg (k = 1)] followed by the United States of America (k = 11, 27%). Some studies were based in East Asia (k = 11, 25%) including in Hong Kong (k = 7), Korea (k = 1) and Japan (k = 3). Only one study was conducted in Canada. Most studies (k = 33, 82%) had received funding to conduct their work.

## Quality assessment and risk of bias in included studies

The quality assessments revealed that few studies were rated as strong (k = 7), with most rates as moderate (k = 16) or weak (k = 17) (S2 and S3 Tables). Most studies did not adequately report on participant selection, resulting in three-quarters being rated as moderate in selection bias (n = 30, 75%). This trend was comparable across both randomised and non-randomised studies. The poor quality of many studies can also be partly attributed to the confounding variable rating. Reporting on ethnicity, age, family or socioeconomic status of families was often poorly described with limited matching again across most study groups (n = 29, 72%); and again, this was as much a feature of RCTs as non-RCTs. Effort was made to contact authors for

**Table 1. Summary of study characteristics sorted by study design and mostly used robot.**

| Reference; Country; Funding | Robot group | Control group | IQ | Robot type | Adverse events | Session details | Risk of bias (overall) | Measure | Outcome |
|---|---|---|---|---|---|---|---|---|---|
| | | | | RANDOMISED CONTROLLED TRIALS | | | | | |
| Huskens et al., 2013; Netherlands; funded [48] | N = 3 (100% male); 8–12 years old | Human therapist; N = 3 (100% male); 8–12 years old | 85–111 | NAO–Humanoid robot | Not reported | 5 sessions; 30 minutes; Clinic room | Moderate | Video recording | No significant between-group differences in self-initiated questions at 19-21-week assessment. |
| Marino et al., 2020 Italy; funded [45] | N = 7 (86% male); 4–8 years old; Italian | Human therapist; N = 7 (86% male); 4–8 years old; Italian | 82–121 | NAO–Humanoid robot | Not reported | 12 sessions; 90 minutes; Laboratory | Strong | Test of Emotional Comprehension (TEC) & Emotional Lexicon Test (ELT) | Improved emotional recognition and comprehension in robot group at 12-week assessment. |
| So et al., 2018; Hong Kong; funded [38] | N = 7 (71% male); 6–12 years old; Chinese | Waitlist group robot sessions after research completion; N = 6 (83% Males); 6–12 years old; Chinese | 49–67 | NAO–Humanoid robot | Not reported | 24 sessions; 30 minutes; School | Weak | Video recording | Improved motor imitation (e.g., gestural accuracy) at 12-week assessment for robot group. |
| So et al., 2018; Hong Kong; funded [39] | N = 15 (87% male); 4–6 years old; Chinese | Waitlist group robot sessions after research completion; N = 15 (93% Males); 4–6 years old; Chinese | Not reported | NAO–Humanoid robot | Not reported | 8 sessions; 30 minutes; School | Weak | Video recording | Improved motor imitation (e.g., gestural accuracy) at 10-week assessment for robot group. |
| So et al., 2019; Hong Kong; funded [40] | N = 13 (85% male); 4–6 years old; Chinese | Waitlist group robot sessions after research completion; N = 11 (93% male); 4–6 years old; Chinese | Not reported | NAO–Humanoid robot | Not reported | 12 sessions; 45 minutes; Clinic room | Weak | Video recording | Improved narrative skills at 12-week assessment for robot group. |
| So et al., 2019; Hong Kong; funded [41] | N = 12 (83% male); 6–12 years old; Chinese | Human therapist; N = 11 (91% male); 6–12 years old; Chinese | 46–74 | NAO–Humanoid robot | Not reported | 4–5 sessions; 30 minutes; School | Weak | Video recording | No significant between-group differences in motor imitation (e.g., gestural accuracy) at 10-week assessment. |
| So et al., 2020; Hong Kong; funded [40] | N = 12 (83% male); 4–6 years old; Chinese | Waitlist group robot sessions after research completion N = 11 (91% male); 4–6 years old; Chinese | Not reported | NAO–Humanoid robot | Not reported | 9 sessions; 45 minutes; Clinic room | Weak | Video recording | Improved joint attention in robot-based drama sessions at 9-week assessment. |

(*Continued*)

**Table 1.** (Continued)

| Reference; Country; Funding | Robot group | Control group | IQ | Robot type | Adverse events | Session details | Risk of bias (overall) | Measure | Outcome |
|---|---|---|---|---|---|---|---|---|---|
| Srinivasan et al., 2015; USA; funded [20] | N = 12 (92% male); 5–12 years old; Caucasian, African American, Asian Hispanic, Mixed African American and Caucasian, Mixed Caucasian and Hispanic | Human therapist; N = 12 (83% male); Comparison group (tabletop activities) N = 12 (83% male); 5–12 years old; Caucasian, African American, Asian Hispanic, Mixed African American and Caucasian, Mixed Caucasian and Hispanic | Not reported | NAO–Humanoid robot & Rovio robot | Not reported | 32 sessions; 15 minutes; Home | Moderate | Video recording | Improved gestural imitation at 10-week assessment in robot group. |
| Srinivasan et al., 2015; USA; funded [21] (overlapping sample) | N = 12 (92% male); 5–12 years old; Caucasian, African American, Asian Hispanic, Mixed African American and Caucasian, Mixed Caucasian and Hispanic | Human therapist; N = 12 (92% male); Comparison group (tabletop activities); 5–12 years old; Caucasian, African American, Asian Hispanic, Mixed African American and Caucasian, Mixed Caucasian and Hispanic | Not reported | NAO–Humanoid robot & Rovio robot | Not reported | 32 sessions; 15 minutes; Home | Weak | Video recording | Improved repetitive behaviour at 10-week assessment for human therapist group. |
| Srinivasan et al., 2016; USA; funded [37] (overlapping sample) | N = 12 (92% male); 5–12 years old; Caucasian, African American, Asian Hispanic, Mixed African American and Caucasian, Mixed Caucasian and Hispanic | Human therapist; N = 12 (83% male); Comparison group (tabletop activities) N = 12 (88% male); 5–12 years old; Caucasian, African American, Asian Hispanic, Mixed African American and Caucasian, Mixed Caucasian and Hispanic | Not reported | NAO–Humanoid robot & Rovio robot | Not reported | 32 sessions; 45 minutes; Home | Moderate | Video recording | Improved social skills at 10-week assessment in robot group. |

(Continued)

**Table 1.** (Continued)

| Reference; Country; Funding | Robot group | Control group | IQ | Robot type | Adverse events | Session details | Risk of bias (overall) | Measure | Outcome |
|---|---|---|---|---|---|---|---|---|---|
| Srinivasan et al., 2016; USA; funded [37] (overlapping sample) | N = 12 (92% male); 5–12 years old; Caucasian, African American, Asian Hispanic, Mixed African American and Caucasian, Mixed Caucasian and Hispanic | Human therapist; N = 12 (83% male); Comparison group (tabletop activities) N = 12 (88% male); 5–12 years old; Caucasian, African American, Asian Hispanic, Mixed African American and Caucasian, Mixed Caucasian and Hispanic | Not reported | NAO–Humanoid robot & Rovio robot | Not reported | 32 sessions; 45 minutes; Home | Moderate | Video recording | Improved repetitive behaviour at 10-week assessment for human therapist group. |
| Zheng et al., 2020; USA; funded [54] | N = 11 (gender not reported); 1.64–3.14 years old | Waitlist group robot sessions after research completion N = 9 (gender not reported); 1.64–3.14 years old | Mean = 58.81 | NAO–Humanoid robot | Two children in waitlist and one child in robot group left at first session due to distress | 4 sessions; 10 minutes; Clinic room | Weak | Video recording | No difference in joint attention skills at 9-week assessment. |
| De Korte et al., 2020; Netherlands; funded [51] | N = 24 (83% male); 3–8 years old | Parent Pivotal Response Treatment N = 20 (85% male); 3–8 years old | Not reported | NAO–Humanoid robot | Not reported | 20 sessions; 45 minutes; Home | Strong | Video recording | Improved self-initiation in robot-assisted Pivotal Response Treatment at 3-month assessment. |
| So et al., 2020; Hong Kong; funded [47] | N = 18 –Tier 1 (N = 6 (67% male), Tier 2 N = 6 (100% male), Tier 3 (N = 6; 100% male); Tier 1 received the intervention earlier than Tiers 2 and 3); all aged 6–8 years old; Chinese | Not applicable | <70 | HUMANE–Humanoid robot | Not reported | 6 sessions; 30 minutes; School | Moderate | Video recording | Improved joint attention at 4–8 weeks assessment in all Tiers. |
| Yun et al., 2017; Korea; funded [67] | N = 8(100% male); 4–7 years old | Human therapist; N = 7 (100% male); 4–7 years old | >60 | iRobiQ & CARO–Humanoid robot | None | 8 sessions; 30–40 minutes; Unknown location | Strong | Video recording | No significant between-group differences in eye-contact at 10-week assessment. |

(Continued)

**Table 1.** (Continued)

| Reference; Country; Funding | Robot group | Control group | IQ | Robot type | Adverse events | Session details | Risk of bias (overall) | Measure | Outcome |
|---|---|---|---|---|---|---|---|---|---|
| Costescu et al., 2015; Romania; funded [16] | N = 12 (74% male); 6–12 years old | Human therapist; N = 15 (74% male); 6–12 years old | Not reported | Keepon-humanoid snowman robot | Not reported | 6 sessions; 120 minutes; School | Moderate | Video recording | Improved emotional intensity and reduced frequency of irrational beliefs in robot group. |
| Pop et al., 2013; Romania; funded [55] | N = 7 (100% male); 4–9 years old | Computer-based session; N = 6/ control group no intervention; N = 7; (100% male); 4–9 years old | Not reported | Probo–Mammoth robot | Not reported | 1 session; 10–15 minutes; Clinic room | Strong | Video recording and 7-point Likert scale | Decreased level of prompt in robot group. |
| Pop et al., 2014; Romania; funded [57] | N = 5 (100% male); 4–7 years old | Human therapist; N = 6 (100% male); 4–7 years old | >70 | Probo–Mammoth robot | Not reported | 1 session; unknown duration; Clinic room | Strong | Video recording and 7-point Likert scale | Improved level of engagement in robot group. |
| Simut et al., 2016; Belgium; funded [59] | N = 30 (90% male); 5–8 years old | Human therapist; N = 30 (90% male); 5–8 years old | 70–119 | Probo–Mammoth robot | Not reported | 1 session, 15 minutes; School | Moderate | Video recording | No significant between-group differences in social skills (e.g., eye-contact, joint attention). |
| Kim et al., 2013; USA; funded [43] | N = 24 (87% male); 4–12 years old; white, two were black and two were Hispanic or Latino | Human therapist; N = 24 (87% male); 4–12 years old; white, two were black and two were Hispanic or Latino | 72–119 | Pleo–Dinosaur robot | Not reported | 1 session; 6 minutes; Clinic room | Moderate | Video recording | No significant between-group differences in number of utterances. |
| Kim et al., 2015; USA; funded [36] (overlapping sample) | N = 24 (87% male); 4–12 years old | Human therapist; N = 24 (87% male); 4–12 years old | 72–119 | Pleo–Dinosaur robot | Not reported | 1 session; 6 minutes; Clinic room | Moderate | Video recording | Improved level of enjoyment and number of words in robot group. |
| **NON-RANDOMISED CONTROLLED TRIALS** | | | | | | | | | |
| Huskens et al., 2015; USA; funded [49] | N = 3 pairs of 1ASD:1sibling (67% male); 5–10 years old | Not applicable | >80 | NAO–Humanoid robot | Aggression to sibling | 6–8 sessions; 30 minutes; Clinic room | Moderate | Video recording | No significant difference in collaborative behaviour at 12-week assessment. |
| Kaboski et al., 2015; USA; funded [50] | N = 8 pairs of 1ASD:1TD (100% male); 12–17 years old | Not applicable | Mean ASD = 106 Mean TD = 112 | NAO–Humanoid robot | Not reported | 5 sessions; 180 minutes; Robotic camp | Strong | Social Anxiety Scale for Children-Revised (SASCR), Social Anxiety Scale Adolescents (SAS-A), Social Skills Improvement System (SSIS) | Partial success. Significant decrease in social anxiety for ASD group only. No significant changes in social skills for both groups at 2-week assessment. |

*(Continued)*

**Table 1.** (*Continued*)

| Reference; Country; Funding | Robot group | Control group | IQ | Robot type | Adverse events | Session details | Risk of bias (overall) | Measure | Outcome |
|---|---|---|---|---|---|---|---|---|---|
| So et al., 2016; Hong Kong; not reported [46] | N = 20 (75% male); 6–12 years old; Chinese; | Not applicable | 51–72 | NAO–Humanoid robot | Not reported | 8 sessions; 30 minutes; School | Weak | Unclear | Improved motor imitation (e.g., gestural accuracy) at 12–14 week assessment for robot group. |
| Tapus et al., 2012; Romania; not reported [52] | N = 4 (100% Male); 2–6 years old | Human therapist; N = 4 (100% Male); 2–6 years old | Not reported | NAO–Humanoid robot | Not reported | 23–26 sessions; 2–5 minutes with 10minutes break; unclear duration; Clinic room | Moderate | Video recording | Partial success. Mixed results for eye-contact, initiations, attention between groups at 4-week assessment. Individual data presented per child. |
| Warren et al., 2015; USA; funded [53] | N = 6 (100% male); 2.5–4 years old | Not applicable | Not reported | NAO–Humanoid robot | Not reported | 4 sessions; unclear duration; Laboratory | Weak | Video recording | Improved attention at 2-week assessment. |
| Zheng et al., 2016; USA; not reported [44] | N = 6 (100% male); 2.5–4 years old; Caucasian | Not applicable | Not reported | NAO–Humanoid robot | Not reported | 6 sessions; unclear duration; Laboratory | Weak | Video recording | The robot attracted the attention at 8-month assessment. |
| Kumazaki et al., 2018; Japan; funded [68] | N = 11 (82% male); mean age = 15.91 | Human therapist; N = 11 (82% male) mean age = 15.91 | Not reported | ACTROID-F & CommU–Humanoid robot | Not reported | 1 session; 5 minutes; Clinic room | Moderate | Audio recording | Improved in length self-disclosure statements in CommU (simple) robot group. |
| Kumazaki et al., 2018b; Japan; funded [82] | N = 16 (75% male); 5–6 years old | Human therapist; N = 12 (58% male); 5–6 years old | >70 | CommU–Humanoid robot | one child in robot group distressed–unable to complete session | 1 session; 15 minutes; Unknown location | Moderate | Video recording | Improved joint attention in robot group. |
| Yoshikawa et al., 2019; Japan; funded [70] | N = 4 (100% male); 15–18 years old | Human therapist; N = 4 (100% male); 15–18 years old | Not reported | Actroid-F–Humanoid robot | Not reported | 5 sessions; one day; Laboratory | Weak | Video recording & eye tracker | Improved eye-contact in robot group. |
| Srinivasan et al., 2013; USA; not reported [71] | N = 1 (100% male); 7 years old | Child-led condition; N = 1 (100% male); 7 years old | Not reported | Isobot–Humanoid robot | Not reported | 8 sessions; 30 minutes; Unknown location | Moderate | Video recording; Sensory Integration and Praxis Test (SIPT) | Improved motor imitation skills in robot group at 6-week assessment. |
| Srinivasan & Bhat, 2014; USA; funded [66] | N = 2 (100% male); 7–8 years old | Not applicable | Not reported | Isobot–Humanoid robot | Not reported | 8 sessions; 30 minutes; Home | Moderate | Video recording | Decreasing attention at 6-week assessment. |
| Costa et al., 2018; Luxembourg; funded [19] | N = 15 (100% male); 4–14 years old | Human therapist; N = 15 (100% male); 4–14 years old | 80–120 | Qtrobot–Humanoid robot | Not reported | 1 session; 1.5–4 minutes; Human vs robot; Laboratory | Moderate | Video recording | Improved attention and repetitive behaviours in robot group. |

(*Continued*)

**Table 1.** (Continued)

| Reference; Country; Funding | Robot group | Control group | IQ | Robot type | Adverse events | Session details | Risk of bias (overall) | Measure | Outcome |
|---|---|---|---|---|---|---|---|---|---|
| Duquette et al., 2008; Canada; funded [14] | N = 2 (100% male); 4–5 years old | Human therapist; N = 2 (50% male); 5 years old | Not reported | Tito–humanoid robot | Not reported | 22 sessions; 3–4 minutes; Laboratory | Weak | Video recording | Partial success. Mixed findings in imitation (e.g., verbal, motor, facial) skills between groups at 7-week assessment. |
| Scassellati et al., 2018; USA; funded [69] | N = 12 (58% male); 6–12 years old | Not applicable | >70 | No name–Humanoid robot | Not reported | 30 sessions; 30 minutes; Home | Weak | Video and audio recor6ding | Improved social skills (e.g., initiations, joint attention eye-contact, engagement) at 4-week assessment. |
| Pop et al., 2013; Romania; funded [56] | N = 3 (100% male); 5–6 years old | Not applicable | Not reported | Probo—Mammoth robot | Not reported | 1 session; Clinic room | Strong | Video recording and qualitative notes | Improved emotional recognition. |
| Simut et al., 2012; Romania; funded [58] | N = 4 (50% male); 4–9 years old | Human therapist; N = 4 (50% male); 4–9 years old | Not reported | Probo—Mammoth robot | Not reported | 6 sessions; 15 minutes; Clinic room | Moderate | 7-point Likert scale | Decreased level of prompt in robot group. |
| Vanderborght et al., 2012; Romania; funded [60] | N = 4 (50% male); 4–9 years old | Human therapist; N = 4 (50% male); 4–9 years old | Not reported | Probo—Mammoth robot | Not reported | 6–8 sessions; 10–20 minutes; Clinic room | Moderate | Video recording | Decreased level of prompt in robot group at 4-week assessment. |
| Silva et al., 2018 Portugal; not reported [61] | N = 10 (100% male); 6–9 years old | Living dog; N = 10 (100% male); 6–9 years old | Not reported | Zoomer–Dog robot | Not reported | 3 sessions; 10 minutes; Home | Weak | Video recording | Improved emotional regulation in living dog condition at 4-week assessment. |
| Silva et al., 2019 Portugal; funded [18] | N = 10 (100% male); 6–9 years old | Living dog; N = 10 (100% male); 6–9 years old | Not reported | Zoomer–Dog robot | Not reported | 1 session; 3 minutes; Home | Weak | Video recording | Improved emotional regulation and social communication in living dog condition at 4-week assessment. |
| Silva et al., 2020; Portugal; funded [62] | N = 10 (100% male); 5–8 years old | Living dog N = 10 (100% male); 5–8 years old | Not reported | Zoomer–Dog robot | Not reported | 1 session; not reported minutes; Home | Moderate | Video recording | Improved imitation in living dog condition. |
| Puyon & Giannopulu, 2013 France; not reported [62] | Game group; N = 11 (72% male); 7–8 years old | No game group; N = 11 (72% male); 7–8 years old | Not reported | "POL"–chicken robot | Not reported | 1 session; 10 minutes; Clinic room | Weak | Video recording | Improved eye-contact, number of words, better posture in game play robot condition. |

(Continued)

**Table 1.** (Continued)

| Reference; Country; Funding | Robot group | Control group | IQ | Robot type | Adverse events | Session details | Risk of bias (overall) | Measure | Outcome |
|---|---|---|---|---|---|---|---|---|---|
| Pierno et al., 2008 Italy; funded [63] | N = 12 (50% male); aged 10–13 years old | Human therapist; N = 12 (50% male); aged 10–13 years old | Not reported | Robotic arm–industrial robot | Not reported | 1 session; 60 minutes; Laboratory | Weak | Video recording | Improved attention in robot group. |
| Giannopulu et al., 2014 France; not reported [64] | N = 15 (73% male); 6–7 years old | Human therapist; N = 15 (73% male); 6–7 years old | Not reported | "Pekoppa"–other robot | Not reported | 1 session; 15 minutes; Clinic room | Weak | Unclear | Improved expressie language in robot group. |

further information, and five author responses were received, which resulted in information being classified as missing and thus reducing study quality. Finally, outcome assessors (e.g., researchers) were not blinded in (n = 38, 95%) of the studies, including often in RCTs as well as non-RCTs. Promisingly, most studies (n = 36, 90%) used appropriate methods to collect data including video data. Studies provided details about the position of the cameras and the use of at least two coders including inter-rater reliability between coders. Most studies (n = 28, 70%) also reported the number of participants approached, screened, and completed the intervention.

## Robot types

Four robot types were used that can be characterised according to their appearance in the following categories: humanoid, animaloid, and other. A humanoid robot is distinguished by its resemblance to the human body. In general, a humanoid robot has a head, torso, two arms and legs. Some humanoid robots may have facial characteristics including eyes, nose and mouth whereas other humanoids may model part of the body from the waist up. Humanoid robots were used in 67% (27 out of 40) of the included studies. The robot platforms that facilitated a session with children and young people on the autism spectrum were the following: NAO, QTrobot, CommU, ACTROID-F, Isobot, Tito, iRobiQ, Caro, Keepon and HUMANE. The most frequently used robot was NAO which was used in 17 studies [20, 38–42, 44–54]. Studies used humanoid robots to examine a range of skills including eye-contact, imitation, joint attention, social skills and emotional regulation.

The use of animaloid (or animal-like) robots, such as an elephant, chicken, dinosaur and dogs, was examined in 11 out of 40 studies (27%). In the review, the *Probo* robot (elephant-like) was referenced in six studies [55–60]. Other animaloid robots were the robot, *Pleo*, [36], the dog robot *Zoomer*, [17, 18, 61] and *POL* (chicken-like) [62]. These robots facilitated sessions focusing on eye contact, imitation, joint attention and social skills.

The remaining 'other' robot category included a robotic arm and a plant robot. The robotic arm was used with children and young people on the autism spectrum to examine imitation and eye-contact [63]. The plant robot, called *'Pekoppa'*, was fully programmable with integrated sensors that allowed the robot to model a range of functions. Pekoppa was used with neurotypical children and young people and children and young people on the autism spectrum to compare the differences in heart rate, verbal fluency, and emotional response [64].

When exploring trends across study type, it appeared that humanoid robots were used in both RCTs (n = 12) and non-RCTs (n = 14). In particular, the robot NAO was utilised in 10

RCTs compared to six non-RCTs (Table 1). Non-RCTs were therefore more likely to include a broader range of robot platforms.

## Settings

Intervention sessions with robots took place in five different settings. The most common location was autism centres/clinic rooms (n = 15) followed by home (n = 7), school (n = 7) and laboratories (n = 7). RCTs showed a trend to be more likely to take place in autism centres/clinics (n = 7 out of 17; 41% versus n = 8 or in a familiar environment such as school (n = 6 out of 17; 35% versus n = 1 out of 23; 4%). Sessions at home were more common in non-RCTs (n = 5; 22%) compared to RCTs (n = 2; 12%). Similarly, sessions in a laboratory were more common in non-RCTs (n = 6; 26%) than RCTs (n = 1; 5%) (S5 Table).

Sixty-five percent (n = 11) of RCTs and 78% (n = 18) of non-RCTs indicated a positive benefit of intervention. The mean duration of robot-intervention was 8.45 sessions and each session lasting an average of 33.5 minutes—most likely occurring once (22%) or twice weekly (28%) over the intervention period, though there was considerable variability (Table 2). The first session was usually a familiarisation meeting with the child and the robot/play partner. Intervention sessions tended to be longer in RCTs, with a mean of 35 (range: 6–120) minutes versus 27 (range: 3–180) minutes in non-RCTs. Similarly, the number of sessions was greater in RCTs with a mean of 9 sessions (range: 1–32) compared to 7 (range: 1–30) in non-RCTs. Notably, play partners across studies were more often 'professionals' for example, researchers or healthcare workforce (90%). Children and young people on the autism spectrum had individual sessions apart from in three studies [49, 50, 65].

## Robot's role in intervention

During intervention, robots took on the role of a social interface. Hence the robot moved its head and eyes to express emotions via facial expressions (e.g., happy, sad) or verbally, became a storyteller, an imitation agent, an intermediate to attract the eye gaze of the child on the autism spectrum or facilitated collaboration within a small group of two children and young people or an object where the child on the autism spectrum engaged in free play. In most

**Table 2. Summary of features of robot-mediated intervention.**

| Robot session characteristics | |
|---|---|
| Number of sessions Mean (SD; range) | 8.45 (9.52; 1–32 sessions) |
| Duration per session (mins) Mean (SD; range) | 32 (35.85; 3–180 mins) |
| **Session frequency** | **N (%)** |
| Single session | 2 (5%) |
| Daily | 4 (10%) |
| Once a week | 9 (22%) |
| Twice a week | 11 (28%) |
| Three times a week | 1 (2%) |
| Varied frequency | 4 (10%) |
| Not reported | 9 (23%) |
| **Play partner** | **N (%)** |
| Researcher | 26 (65%) |
| Child/Clinical Psychologist/ Psychotherapist | 10 (25%) |
| Parent | 1 (2%) |
| No play partner | 3 (8%) |

studies (n = 38, 95%), children and young people engaged in a triadic relationship with an adult/therapist where the robot acted as a mediator. A typical session involved the play partner controlling the robot via a laptop/computer. Two studies used a fully autonomous robot to play independently without the guidance of an adult partner. The control group in RCTs was often a human therapist engaging the child with the same or similar activities apart from five studies that used a waitlist and so, received the robot session after study completion. It was unclear whether the children in waitlist studies were receiving no treatment interventions as part of their educational and community settings.

## Targeted skills and outcomes

Studies included in this review targeted a number of skills that can be clustered into 3 main categories: (1) social and communication skills [narrative skills (n = 1), self-initiated questions (n = 4), engagement (n = 2), self-disclosure (n = 1), collaborative play (n = 1), level of prompting (n = 3), joint attention (n = 6), eye-contact (n = 6), imitation (n = 7)]; (2) emotional development [recognition and/or understanding (n = 2), emotional regulation (n = 4)]; and (3) motor skills [stereotyped or repetitive behaviour (n = 1)]. Over two-thirds (n = 29; 72%) of the articles reported a positive impact of a robot-mediated intervention in children and young people on the autism spectrum. Less than a quarter of the included articles (n = 10, 25%) reported no difference in targeted skills development. Finally, one article reported a decline in attention skills during the intervention period [66].

The majority of the targeted skills were measured through the examination of video recordings and coding procedures completed by researchers whereas only two studies [45, 50] used standardised assessment tools to examine social skills and emotional comprehension (Table 1). Nine studies [20, 38–42, 45, 55, 67] used blinded researchers to administer the questionnaires and one study [50] relied on parent-reported outcomes. Another three studies [55, 57, 58] utilised child self-report methods, three used qualitative methods (e.g., audio recording, notes by researchers) [56, 68, 69] and two used eye-tracking [70] or a sensory integration and praxis test [71] in conjunction with video recordings. Two studies made explicit reference to the benefits of the robot-mediated intervention at two weeks [48] and four weeks [40] following the end of the intervention. Finally, Marino and colleagues [45] reported that children on the autism spectrum in both groups (robot and human) spontaneously practiced the trained skills addressing generalisation issues. No studies included evaluation of health economics related to intervention delivery.

## Meta-analysis

Hedge's g was calculated for RCTs examining outcomes relating to: social (k = 7), emotional (k = 2) and motor (k = 3) abilities and for all three areas combined (k = 12). This provided a total of 346 participants (175 assigned to robot and 171 assigned to control conditions). The control condition of the included studies comprised of children in a human therapist group apart from three studies [39, 41, 42] that had a waitlist group (where the children received the robot session after study completion).

Nine RCTs were excluded from the analyses because of: (1) overlapping samples [20, 21, 36, 37]; and (2) use of waitlist group and/or no reporting (or sharing when contacted directly) of means, SDs or effect sizes [38, 42, 54, 55, 57]. The included RCTs had quite good quality ratings: strong (k = 4), moderate (k = 6) and weak (k = 3). All three of the weak ratings were for the studies by So and colleagues [39–41] and all were assessing motor outcomes.

RCTs providing sufficient data for emotion-based outcomes to be examined revealed a nonsignificant effect size (g = 0.63 [95%CI -1.43 to 2.69]; k = 2). Heterogeneity was high

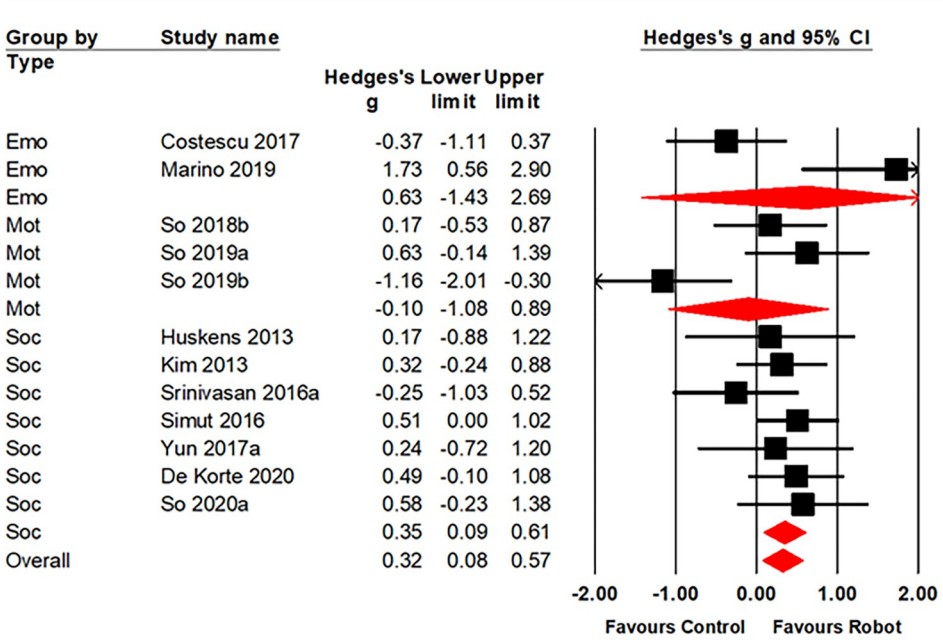

**Fig 2. Forest plot showing efficacy of robot intervention on emotional, motor and social outcome variables.**

(I2 = 88.65). For trials assessing motor outcomes, the effect size was again non-significant (g = -0.10 [95%CI -1.08 to 0.89]; k = 3) and heterogeneity was again high (I2 = 79.63). For social outcomes, the effect size was significant (g = 0.35 [95%CI 0.09 to 0.61; k = 7) and heterogeneity was low (I2 = 0.00). When we combined all three sets of outcomes to assess any pooled benefit of robot-mediated interventions (Fig 2), the effect size was significant (g = 0.33 [95%CI 0.08 to 0.57; k = 12) and heterogeneity was moderate and significant (I2 = 54.48). Visual analysis of funnel plots did not suggest any asymmetry and evidence of obvious publication bias (S1 Fig).

Although there is no definitive minimum number of studies required for meta-regression, we follow the general recommendation of at least 6 to 10 studies for a continuous variable (Higgins et al., 2019). Given this, we used meta-regression to assess possible moderators across all 12 RCTs in the meta-analysis. We found that age was a significant moderator (z = -1.97, df = 12, p = .02) (S2 Fig), with effect sizes being significantly larger in younger samples. Indeed, age accounted for nearly a third of the variance in effect sizes (analog r$^2$ = .32). None of the other continuous variable moderators were significant including: total length of time in sessions (z = 0.40 df = 12, p = .35); proportion of male participants (z = 0.97, df = 12, p = .17); and IQ (z = 1.44 df = 8 p = .07).

We also used sub-group analysis to see if the context (home, school, clinic) impacted effect sizes across all RCTs. This analysis showed a significant impact of robots in the clinic (g = 0.57 (95%0.16 to 0.98; k = 5) with low heterogeneity (I2 = 21.96). By contrast robots were not efficacious in either the home g = 0.16 (-0.56 to 0.89; k = 2; I2 = 55.55) or in school g = -0.16 (-0.85 to 0.53; k = 4, I2 = 75.19), though few trials examined the latter.

## Discussion

The current systematic review summarises evidence on the use of social robots with children and young people on the autism spectrum. We aimed to examine the typology of robots, the

settings of robot-mediated interventions, the function of the robot during intervention and the specific skills targeted for therapeutic gain. Notably, the current review provides the first meta-analysis to estimate the efficacy of robots to bring about meaningful gains, particularly in social processing. We also highlight key moderators of effect sizes–these include age, with younger individuals appearing to benefit more and with effects being significantly larger in the clinic compared to school. Additionally, we found that effect sizes were not moderated by the length of therapeutic intervention. This suggests that relatively brief forms of robot-based interventions might be helpful (most protocols include an average of 8 sessions each lasting approximately 30 minutes), with meaningful improvement in social communication at least in the immediate post-intervention period. Longer-term follow-up was lacking and limits the extent to which we can generalize learned skills to daily life. For example, of the RCTs included in the review, the median duration of follow-up data reported was 10 weeks since baseline assessment. To understand the generalization of skills, and both to support commissioning decisions and to set parent expectations about intervention outcomes, longer-term data will be essential.

Most studies in this review focused on evaluating humanoid robotic platforms, with a small minority assessing animaloid types. In line with the social nature of the robots, the most common outcomes could be grouped into three clusters, with most studies focusing on social communication, emotional outcomes, and motor imitation. Most often outcomes were assessed using video data over other assessment types. However, in studies using video-recording data, raters were not blinded in 95%, typically because the data being coded was during the intervention when the robot is clearly visible. To overcome this bias, we advance that future trials use naturalistic observations (e.g., free play) to better assess the extent of skills generalisation and by raters who are blinded to the intervention the child/young person received.

The most common intervention settings were autism centres/clinics, with fewer studies based in other everyday learning environments such as the child's school and home. In this context, a recent qualitative study found that educators are not 'uncritically approving' of the use of social robots in schools [72]. Though robots are recognised as being motivating, engaging, predicable, and consistent, educators have called for clear protocols on how and why robots are being used to ensure effective learning support [72]. Few school-based RCTs could be included in our meta-analysis and so, the non-significant effect is unsurprising given the heterogeneity and lack of power. Future research needs to optimise intervention protocols and their practicability in educational settings, as well as effective training and on-going support for the educators involved. Though autism centres/clinics currently yield the clearest therapeutic benefit, everyday settings potentially offer more feasible routes to embed intervention and may be more cost-effective, though health economic evaluation data are currently lacking. Further, community settings have been identified as underutilised in autism research and offer a naturalistic environment where a range of expertise can be harnessed for the shared goal of improving everyday functioning for both children and families [73]. Effectively integrating robots in a range of settings remains crucial to translate research into practice.

In 80% of the studies, the robots acted to mediate between the child and play partner, who typically controlled the robot through a keypad. Although some robots were designed with the architecture of being autonomous (e.g., NAO), human therapists use robots as focal points to engage children and youth on the autism spectrum in a session in case they find human interaction challenging [15, 74]. Human therapists also use robots to demonstrate movements and/or facial expressions so that children can mimic these. In this way, robots preserve a human therapist's time and energy [10, 74, 75] whilst also benefitting from consistency as robots are more likely to produce the same movement or expression each time. Unsurprisingly our findings evidence the advantage of social robot-mediated interventions in learning or skill domains that map onto social communication. The meta-analytic evidence here provides

support for a small-moderate benefit of robots on social-related outcomes; however, specific impact on emotional and motor outcomes remains elusive because of the small numbers of trials, the small samples per trial and the high heterogeneity currently associated with those outcomes–all of which reduce power to detect efficacy in the emotion and motor domains.

Turning to risk of bias assessment, most studies were rated as either weak or moderate using the Quality Assessment Tool for Quantitative Studies [35]. Only seven (17%) studies were rated as strong, and five of these were RCTs, where higher standards of rigour and reporting might be expected. Common weaknesses related to selection bias [k = 36 (90%); 15 RCTs; 21 non-RCTs] and the reporting on confounding variables [k = 35 (87%); 17 RCTs; 20 non-RCTs]. Other researchers have also commented on selection bias being common in autism research, particularly the exclusion of youth with a diagnosis of autism and intellectual disability [76]. Similarly, many studies did not assess intelligence. The current meta-analysis found a trend toward larger effect sizes in samples with higher IQ; however, the meta-analysis had data missing from four RCTs and the samples were somewhat bimodal with two studies having a mean IQs in the 5–60 range, while the remainder were 90 to 105. Transparency and consistency in reporting sample characteristics is therefore essential in future research. This will help delineate for whom robot-mediated interventions are more effective, therefore allowing better targeting and adaptation of intervention protocols.

Finally, the increase in research studies in the field of autism and robot-mediated intervention in the past 15 years and the proportion of studies that have received funding clearly evidences the growth in interest in this form of therapy for autistic individuals, mostly in the United States and Europe. The relative lack of research on autism and social robotics from other countries may however signal a need for more global perspectives on human-robot interaction and cultural influences on autism intervention design [77]. For example, Hashim & Yussof [77] suggest that robots could be humanised more to support ethical, spiritual and religious learning, acting to increase cross-cultural appeal in autism research. Given the outcomes of our meta-analyses, we encourage approaches that seek to adapt interventions for cross-cultural benefit.

## Strengths and limitations

As far as we are aware, the current review is both the first to be preregistered and the first to meta-analyse some evidence on robot-mediated intervention for autistic children and young people. In doing so, we have provided novel insights absent from other reviews [7, 32]. Our search of the grey literature generated articles that are included in this review and so, offers a more balanced picture of the current literature, minimising potential for publication bias. Evidence from the meta-analysis did not point to any obvious publication bias amongst RCTs. Nonetheless, the number of RCTs that provided sufficient data for meta-analysis were small and the findings should be interpreted in that context. Nonetheless, we believe this new evidence will be helpful to researchers, trialists, clinicians, educators, and parents, especially as the field of technology-assisted learning in autism is seeing an expansion [78–81]. Other limitations might be the large number of articles (k = 102) we excluded owing to their poor reporting on autism diagnosis. Systematically excluding these studies was in line with our preregistered inclusion criteria but may have excluded data from the true population of interest. Studies might however benefit from using clear diagnostic criteria. Further, the evidence from meta-analysis should be considered with the caveat that we could derive data from only 12 RCTs, although we did manage to capture data from almost three quarters of the trials (12/17). We also note that our examination of moderator variables was limited to analyses that were pooled across all three outcomes (social, emotional and motor) in order to obtain

sufficient data points. Despite such limitations, the findings for social outcomes look especially promising, while those relating to emotional and motor abilities require further studies. We were unable to report on adverse events as these data have been poorly reported in studies. Only four studies reported adverse events [49, 54, 67, 82]. Finally, as commented on earlier, the studies included in this review have reported on the outcome of brief exposure and its impact at best in the immediate post intervention or short follow-up period. As such, though robot mediated interventions appear to be promising, the extent to which skills are generalised into everyday life and sustained is unclear.

### Future recommendations

The outcomes of the review suggest the emergence of considerable interest in evaluating the therapeutic benefits of social robots for children and young people on the autism spectrum. Robot (humanoid and animaloid) platforms are suggested as suitable to personalise, scalable and economical and so offer immense opportunity as a form of autism intervention [83, 84]. For intervention benefits to be maximised, however, better reporting across study designs on sample recruitment and characteristics and adverse events, as well as further standardisation of outcome measures is needed. Further, clinical utility will remain limited without evaluation from randomised designs that assess the evidence of immediate as well as more sustained treatment gains. It has been emphasised that any intervention should be implemented consistently across settings and multiple professionals working together to support children and young people on the autism spectrum will increase the likelihood of more rapid progress [85]. Design and evaluation of robotic research would benefit from a multi-disciplinary approach to harness technological developments, methodological considerations, and evaluation of behavioural and psychological outcomes. Further, building consensus across the social robotics research community about intervention evaluation would be advantageous and can draw on existing approaches that have informed similar frameworks in other areas of autism intervention [86].

### Conclusions

Humanoid robots are the most common form of robot intervention employed with children and young people on the autism spectrum. Intervention protocols tend to be brief, and usually implemented in autism centres/clinics, home or to a lesser extent, at school, where robots typically take on the role of a therapeutic mediator. Evidence from the current meta-analyses suggests that effects are larger when trials have been conducted in the clinic rather than at home or in schools, and for younger children suggesting better developmental match. Current research findings however should be interpreted cautiously given the lack of high-quality RCT evidence. To increase assessment of clinical effectiveness, this review identifies a need for more research based on experimental designs and with transparent reporting on sample selection, characteristics, and adverse events, as well as assessment of intervention gains beyond the immediate study period.

### Supporting information

**S1 Fig. Funnel plot exploring publication bias.**
(TIF)

**S2 Fig. The impact of age on robot interventions.**
(TIF)

**S1 Table. PRISMA 2020 checklist.**
(DOCX)

**S2 Table. Search terms per bibliographic database.**
(DOCX)

**S3 Table. Quality assessment for included studies.**
(DOCX)

**S4 Table. Individual study quality assessment overview.**
(DOCX)

**S5 Table. Vote count outcomes by setting.**
(DOCX)

## Author Contributions

**Conceptualization:** Athanasia Kouroupa, Karen Irvine, Silvana E. Mengoni, Shivani Sharma.

**Data curation:** Athanasia Kouroupa.

**Formal analysis:** Athanasia Kouroupa, Keith R. Laws.

**Methodology:** Athanasia Kouroupa, Keith R. Laws, Alister Baird.

**Project administration:** Athanasia Kouroupa.

**Software:** Keith R. Laws.

**Supervision:** Karen Irvine, Silvana E. Mengoni, Shivani Sharma.

**Validation:** Athanasia Kouroupa, Keith R. Laws.

**Visualization:** Athanasia Kouroupa, Keith R. Laws.

**Writing – original draft:** Athanasia Kouroupa.

**Writing – review & editing:** Athanasia Kouroupa, Keith R. Laws, Karen Irvine, Silvana E. Mengoni, Alister Baird, Shivani Sharma.

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
