## [Decision Letter · Decision Letter 0]

31 May 2022

The use of social robots with children and young people on the autism spectrum: A systematic review and meta-analysis

PONE-D-22-06805

Dear Dr. Kouroupa,

We’re pleased to inform you that your manuscript has been judged scientifically suitable for publication and will be formally accepted for publication once it meets all outstanding technical requirements.

Kind regards,

Cristina Vassalle

Academic Editor

PLOS ONE

Additional Editor Comments (optional):

Reviewers' comments:

Reviewer's Responses to Questions

**Comments to the Author**

1. Is the manuscript technically sound, and do the data support the conclusions?

Reviewer #1: Yes

Reviewer #2: Yes

2. Has the statistical analysis been performed appropriately and rigorously? 

Reviewer #1: Yes

Reviewer #2: N/A

3. Have the authors made all data underlying the findings in their manuscript fully available?

Reviewer #1: Yes

Reviewer #2: Yes

4. Is the manuscript presented in an intelligible fashion and written in standard English?

Reviewer #1: Yes

Reviewer #2: Yes

5. Review Comments to the Author

Reviewer #1: Dear Authors, your review is up-to-date and focuses on an extremely topical issue in the field of autism. I applaud you for the rigorousness of your methodology. The results are clear and well described. The conclusions help the reader to take the sometimes overly optimistic results with caution. Too few RCT studies are yet available to draw robust conclusions. The article is well done and for this reason in my opinion it is acceptable for publication.

Reviewer #2: The study is very interesting and provides very insightful information about the use of social robots in working with children and adolescents on the autism spectrum. I would like to emphasize the effort that had to be put into the preparation of this publication. The publication type declared by authors in the system is research article I suggest changing it to review.

In the introduction the authors sufficiently present the research problem and clearly specify the research objective. The systematic review and meta-analysis prepared by the authors were conducted according to PRISMA (Preferred Reporting Items for Systematic Reviews and Meta-Analysis) principles. The method chapter provides the necessary information, the criteria for inclusion and exclusion of works for review are described in detail and correctly. The results of the analysis conducted and the discussion are described in a very clear manner. The authors presented the strengths of the study and described the limitations they were aware of in preparing the paper. The conclusions were presented in a concise and correct manner.

In my opinion, the article can be published in its current form.

6. PLOS authors have the option to publish the peer review history of their article (what does this mean?). If published, this will include your full peer review and any attached files.

Reviewer #1: **Yes: **antonio narzisi

Reviewer #2: No

---

## [Editor Report · Acceptance letter]

6 Jun 2022

PONE-D-22-06805 

The use of social robots with children and young people on the autism spectrum: A systematic review and meta-analysis 

Dear Dr. Kouroupa:

I'm pleased to inform you that your manuscript has been deemed suitable for publication in PLOS ONE. Congratulations! Your manuscript is now with our production department. 

Kind regards, 

on behalf of

Dr. Cristina Vassalle 

Academic Editor

PLOS ONE